# Role of Melatonin and Nitrogen Metabolism in Plants: Implications under Nitrogen-Excess or Nitrogen-Low

**DOI:** 10.3390/ijms232315217

**Published:** 2022-12-02

**Authors:** Marino B. Arnao, Josefa Hernández-Ruiz, Antonio Cano

**Affiliations:** Phytohormones and Plant Development Laboratory, Department of Plant Biology (Plant Physiology), Faculty of Biology, University of Murcia, 30100 Murcia, Spain

**Keywords:** low nitrogen use, melatonin, nitrogen uptake, nitrogen metabolism, NUE, plant growth, plant nutrition, plant stress

## Abstract

Melatonin is a new plant hormone involved in multiple physiological functions in plants such as germination, photosynthesis, plant growth, flowering, fruiting, and senescence, among others. Its protective role in different stress situations, both biotic and abiotic, has been widely demonstrated. Melatonin regulates several routes in primary and secondary plant metabolism through the up/down-regulation of many enzyme/factor genes. Many of the steps of nitrogen metabolism in plants are also regulated by melatonin and are presented in this review. In addition, the ability of melatonin to enhance nitrogen uptake under nitrogen-excess or nitrogen-low conditions is analyzed. A model that summarizes the distribution of nitrogen compounds, and the osmoregulation and redox network responses mediated by melatonin, are presented. The possibilities of using melatonin in crops for more efficient uptake, the assimilation and metabolization of nitrogen from soil, and the implications for Nitrogen Use Efficiency strategies to improve crop yield are also discussed.

## 1. Introduction

Nitrogen is the main element that limits plant productivity in crops, with nitrate being the main nitrogen form for plants. The massive use of nitrates and other nitrogen compounds, such as ammonium, has led to serious problems in agricultural soils, such as the high salinity and contamination of aquifers with nitrogen [1,2,3,4]. Excess nitrogen in soil usually has negative consequences on plant physiology, such as a lower photosynthetic rate, osmotic stress, nitrogen metabolism disorders, and the excessive appearance of ROS (reactive oxygen species) and RNS (reactive nitrogen species). Furthermore, alterations in the assimilation of other elements such as Ca and Mg cause a lower response in defense against pathogens [5]. Either directly or indirectly, excess nitrates increase ammonium levels in the soil, affecting the overall growth of the plant, both in aerial and root systems [6,7,8,9]. Therefore, overuse of nitrogen can result in decreased crop productivity. Therefore, producing more whilst using less nitrogen, the so-called Nitrogen Use Efficiency (NUE), is a recommended practice [10,11,12]. Currently, the use of “smart fertilizers”, such as coated fertilizers, and the application of plant-growth-promoting rhizobacteria (PGPR) to the soil has ostensibly improved the nutrition of nitrogen in plants [13,14,15]. On the other hand, the deficit of nitrogen in soils usually entails great limitations in plant growth and serious nitrogen and carbon metabolic dysfunctions, reducing the photosynthesis and biosynthesis of amino acids and proteins [16,17]. 

Melatonin (*N*-acetyl-5-methoxytryptamine) is a biogenic amine discovered in 1958 in the pineal gland of cow [18], and later in humans [19,20]. Its properties as a hormone which regulates light/dark cycles and other endogenous rhythms in mammals have been extensively studied. Remarkably, melatonin was identified in plants in 1995 simultaneously by three groups of researchers [21,22,23]. Although initially there were many doubts about the presence of melatonin in plant tissues, today it is one of the most prolific and exciting areas of study within plant physiology [24]. In plants, melatonin has a multi-regulatory role, behaving as a “plant master regulator”, stimulating processes such as seed germination, photosynthesis, growth, and rooting, whilst inhibiting leaf senescence, and regulating fruit ripening and senescence. In addition, melatonin considerably increases tolerance to biotic (bacteria, fungi, and viruses) agents and to abiotic stressors, such as water deficit, extreme temperatures, salinity, contaminants, etc. These aspects are of great interest in its application in crops [25,26,27,28,29,30,31].

In plant metabolism, melatonin regulates many primary metabolism pathways, mainly in carbohydrates (starch and sucrose) [32], lipids, and nitrogen compound routes, and also in secondary metabolism pathways (phenolics, flavonoids, and terpenoids) [24]. The broad functions and regulatory capacities of melatonin in plant and animal cells have been studied and their analogies and differences have been compared [33,34].

In this paper, a review of the role of melatonin in nitrogen metabolism in plants is presented. Its regulatory role in the different metabolic stages was analyzed, as well as its general action in situations of excess and the deficiency of nitrogen. In addition, the possibilities of using melatonin in crops for more efficient uptake, the assimilation and metabolization of nitrogen from soil, and the implications for Nitrogen Use Efficiency (NUE) strategies to improve crop yield were also discussed, i.e., a proposal to increase crop yields with suboptimal levels of nitrogen.

## 2. Methodology

In this study, a systematic literature review is conducted on the role of melatonin in the nitrogen metabolism of plants. The bibliometric analysis was conducted in three stages: (i) defining the keywords; (ii) selecting the database, and (iii) searching relevant articles and analyzing data. Peer-reviewed publications were searched covering the period 1995, the year melatonin was first identified in plants, and 2022. The search was performed in the title and keywords of the publications, selecting English and other languages, and other articles in journals and book chapters, both experimental and review types, which were related to melatonin and nitrogen metabolism in plants. A systematic database search of peer-reviewed articles was conducted using the Science Citation Index Expanded (SCI-Expanded) database of the Web of Science from Thomson Reuters. Additionally, Scopus, Google Scholar, and PubMed databases were used. Our Plant Hormones and Development research group from the University of Murcia has ample experience in melatonin and for 25 years has been generating a database integrated by some 4000 references that include all the works related to melatonin in plants. The results indicated that only 75 references were registered, of which 62 are from the last five years. 

## 3. Melatonin Biosynthesis

In plants, melatonin is synthetized from chorismic acid, which is generated from shikimic acid (a condensation product of phosphoenolpyruvate from glycolysis and erythrose 4-phosphate from the pentose phosphate pathway (Figure 1)). Chorismic acid, a precursor to aromatic amino acids (phenylalanine, tyrosine, and tryptophan), is transformed through the anthranilate/indole pathway to tryptophan [35]. Tryptophan is the origin of the melatonin biosynthesis pathway in both animal and plant cells [33,36,37]. In animals, tryptophan is converted to 5-hydroxytryptophan by TPH, an enzyme that apparently has not been identified in plants. Tryptophan is mainly transformed into tryptamine by TDC present in cytoplasm of plant cells, and then up to serotonin (5-hydroxytryptamine) by T5H (in endoplasmic reticulum) (Figure 1). The transformation of serotonin into melatonin is produced in the chloroplast or cytoplasm depending on the enzyme involved. Thus, serotonin can first be acetylated by SNAT to *N*-acetylserotonin (in the chloroplast) and subsequently hydroxylated to melatonin (in the cytoplasm) by ASMT/COMT. Under conditions of stress or excess serotonin, 5-methoxytryptamine is formed preferentially by the action of ASMT/COMT [38], and finally melatonin is generated by SNAT (in the chloroplast) [39,40,41]. Melatonin is usually hydroxylated at different positions on the indole ring, with 2-hydroxymelatonin being the major catabolite in plants, showing interesting regulatory properties [42,43,44]. 

## 4. Role of Melatonin in Nitrogen Metabolism

The role of melatonin on nitrogen metabolism has been studied under conditions of nitrogen excess and deficiency. Table 1 summarizes several representative examples of the effect of melatonin on nitrogen metabolism in different plant species and conditions. Under nitrogen-excess conditions, melatonin treatments induced a decrease in endogenous nitrogen levels in the form of nitrate and ammonium. In cucumber plants grown with excess nitrate, 100 µM melatonin increased nitrogen tolerance and growth, reorganizing NPK balance and lowering nitrogen damage by reducing nitrate and ammonium levels in seedlings. Furthermore, cucumber melatonin-treated seedlings increased enzyme activities, and nitrate reductase (NR), glutamine synthase (GS), glutamate synthase (GOGAT), and glutamate dehydrogenase (GDH) gene expression, reducing the negative effect of excess nitrate [45]. Also in nitrate excess, melatonin, in co-action with nitric oxide (NO), reversed the excess nitrogen inhibition of root growth, and also up/down-regulated several genes of IAA and ABA metabolism, including melatonin biosynthesis genes [46]. Previously Zhao et al. (2012), also investigating cucumber, demonstrated for the first time, that melatonin increased high-temperature tolerance, regulating nitrate and ammonium levels and nitrogen-related enzymes [47]. The role that melatonin plays in increasing nitrogen-excess tolerance was also studied in alfalfa plants (Table 1). In this case, melatonin increased nitrogen-excess tolerance through the up-regulation of NR, GS, GOGAT, and GDH enzymes. Furthermore, it reduced the total nitrogen levels (nitrate and ammonium content) and increased the biomass, length, width of leaves, and energy levels (P and ATP) and decreased the Na, K, and Ca mineral contents [48]. Similar results were obtained in soybean plants treated with melatonin in nitrogen-excess [49].

In nitrogen-normal conditions, melatonin-treated maize seedlings were found to increase nitrogen content (nitrate and nitrite) and decrease ammonium content with respect to the control plants. Furthermore, NR, NiR GS, GOGAT, and GDH activities and gene expression were up-regulated in melatonin treatments [52]. Unexpectedly, melatonin is also able to favorably regulate nitrogen levels under deficit conditions. Winter wheat grown in a nitrogen-deficit medium was capable of increasing nitrogen uptake and nitrate contents in melatonin-treated seedlings, increasing shoot and root growth, as well as yield, and possibly improving nitrogen metabolism [51]. Also in soybean plants, the role of melatonin in nitrogen-low conditions has been studied. In nitrate and ammonium deficiency, melatonin improved plant tolerance, increasing the total number of nodules and fixed nitrogen, and up-regulating several nitrogen-related gene expressions (see Table 1), with the result of an increase in the levels of amino acids, proteins, chlorophyll, and also an increase in seed yield [49,55].

Moreover, in different stress conditions, such as drought stress, melatonin improved stress tolerance and nitrogen uptake. It also improved the contents of amino acid, protein, proline, and ureides, whilst up-regulating NR, NiR, NRT, GS, GOGAT, and GDH gene expression, consequently improving the growth and total biomass of soybean plants [56]. The authors suggested that melatonin regulated the assimilation, metabolism and transport of nitrogen, thereby maintaining the carbon/nitrogen balance [48,50,58,59]. 

## 5. Melatonin in Osmoregulation and Redox Network

Many data point to the regulatory roles of melatonin in plants under stress. Figure 2 shows a diagram with several of the enzymes and other factors in pathways regulated by melatonin. Melatonin increases the contents of several metabolites involved in cellular osmoregulation. Sugar-alcohols (polyols) such as sorbitol, mannitol, glycerol, and inositol, and nitrogen-compounds such as proline and glycine-betaine, among others, are clearly increased with melatonin treatments in several stress situations, and the expression of some related enzymes regulated by melatonin has been demonstrated [24,32]. In addition, the role of melatonin as a main redox homeostasis regulator has been demonstrated. A redox network involving several ROS and RNS, as well as related enzymes such as nitric oxide synthase (NOS-like), NR, respiratory burst oxidase (RBOH), ASA-GSH cycle enzymes, and antioxidant enzymes (superoxide dismutases-SOD, catalases, peroxidases, peroxiredoxins) was controlled direct or indirectly by melatonin [24,28,58,60,61].

Ion homeostasis is also regulated by melatonin. Under saline or alkaline growth conditions, ion homeostasis is re-established in melatonin-treated plants through the regulation of many ion transporters, mainly transporters of Na^+^, K^+^, and Cl^−^, and also phosphate, and sulfur [24,27,62,63]. Under saline stress conditions, melatonin improved the K/Na ratio, increasing K uptake and decreasing Na contents in pepper leaves [64]. Similar results have been observed in rice [65] and tomato [66]. In apple trees, melatonin also improved the contents of N, P, K, Ca, Mg, Cu, Zn, and B in melatonin-treated plants compared with those not treated under drought conditions [50]. Thus, melatonin reestablished ion homeostasis under stress conditions, and influenced global mineral nutrition. The role of melatonin in sulfur metabolism is notable. Melatonin was able to revert sulfur deficiency in plants through the up-regulation of genes involved in sulfur transport and metabolism, including several sulfur transporters such as ATP sulphurylase, 5′-adenylylsulfate reductase, sulphite reductase, and *O*-acetylserin-thiol-lyase, thereby improving sulfur uptake and content, which improved redox homeostasis [67]. This effect of melatonin on sulfur metabolism has been demonstrated mainly under stress conditions [68,69].

## 6. Melatonin, Nitrogen, and Implications in Crops

Figure 2 shows the melatonin-regulated points related to nitrogen metabolism. Several nitrogen uptake genes were up-regulated by melatonin, including NRT1-1, and several NRT2 as nitrate transporters, as well as AMT1-2/5/6 and AMT2-1 as ammonium transporters (Table 1, Figure 2). In general, melatonin enhanced mineral uptake, specifically N, P, and S uptake, increasing stress tolerance and also heavy metal tolerance, which has been applied in phytoremediation studies [62]. NR and NiR genes were up-regulated by melatonin under different experimental conditions and in different plant species, resulting in an increase in nitrate and/or ammonium levels, in nitrogen-normal or under nitrogen-low conditions (Table 1). In nitrogen-excess, melatonin regulates the metabolism of nitrogen through metabolic genes such as GS, NADP-GOGAT, Fd-GOGAT, and GDH, mainly resulting in a decrease in nitrate and ammonium contents (Figure 2). The effect of melatonin on nitrogen uptake and metabolism results in increased contents of nitrogenous compounds such as amino acids, ureides, and proline, along with higher levels of proteins. This consequently improves the plants’ growth and tolerance to stresses (Table 1, Figure 2). Melatonin behaves as an excellent plant master regulator [58] and not only acts on nitrogen metabolism, but also improves important pathways such as photosynthesis (light and Calvin reactions), water economy, carbohydrate and lipid metabolism, the Krebs cycle, osmoregulation, the redox network including the ASC-GSH cycle, plant hormone homeostasis, and many more aspects of the primary metabolism of the plant cells, as recently reviewed [24]. In crops, melatonin enhanced growth and stress tolerance resulting in improved biomass and seed/fruit yield. Some examples can be seen in Table 2. 

Liu et al., (2020) noted that low-nitrogen-tolerant wild soybeans were capable of reducing their energy consumption by decreasing the biosynthesis of amino acids. Concurrently, they enhanced the biosynthesis of proline and secondary metabolites to withstand low-nitrogen stress. According to the authors, enhancing the metabolism of shikimic acid was a unique mechanism involved in the low-nitrogen tolerance of low-nitrogen-tolerant wild soybeans [17]. Under low-nitrogen conditions, the presence of melatonin simulated low-nitrogen-tolerant varieties, reordering amino acid biosynthesis and reinforcing redox network and osmoregulatory responses (Table 1, Figure 2). Moreover, melatonin has a great ability to regulate secondary metabolism, especially polyphenol, isoprenoid, and glucosinolate pathways [24,31]. 

In legume crops such as soybean, melatonin promoted the development of symbiotic root nodules, increasing their number and size, enhancing nitrogen, and biomass accumulation [54]. For example, in soybean, under nitrogen-low, nitrogen-normal, and nitrogen-excess conditions, biomass accumulation increased by 9.8%, 14%, and 11.4%, respectively [49]. Melatonin treatments in soybean at the V3 stage (nodule development) were more determinant in the establishment and increase of the number of nodules than at R5 stage (grain filled), indicating that melatonin plays a relevant role in root nodule development. In this line, the activity of N_2_ fixing-bacteria such as nitrogenase in root nodules was promoted by melatonin at the V3 stage, especially under nitrogen-low conditions [55]. Regarding this, some data have been published on the possible beneficial effects of melatonin in rhizosphere microbial community structure [84,85,86,87,88,89], and also on the effect of the simultaneous application of melatonin and *Rhizobium* in plant growth and stress [90,91,92].

## 7. Conclusions and Perspectives

In relation to nitrogen nutrition, melatonin improves growth, survival rates, and stress tolerance. In nitrogen-excess situations, melatonin increased nitrogen-stress tolerance, whilst reducing nitrate/ammonium uptake and up-regulating nitrogen-related genes. This resulted in a rebalancing of nitrogenous compounds and a redirecting of amino acid, proline, and ureide levels. Under nitrogen-low conditions, an improvement in the levels of nitrogen compounds was observed, with a greater absorption of nitrogen and an increase in levels of amino acids, proteins, and chlorophylls. In some cases of abiotic stress, such as drought or high temperatures, melatonin regulated nitrogen-related genes, optimizing osmoregulation response, mineral uptake, and total nitrogen levels in the tissues. In most of the cases studied, melatonin increased growth and plant biomass. In soybean cultivation, melatonin improves yields, possibly due to the stimulation of the number and size of root nodules.

The effect of melatonin on nitrogen metabolism has been presented as an excellent tool to improve yield in crops with a low nitrogen supply, since melatonin optimizes both mineral uptake and the biosynthesis and redistribution of nitrogen compounds, especially under stress conditions. Nitrogen loss caused by leaching into the environment is a serious problem for the agricultural sector due to the undesirable effects of it as a relevant contaminant agent, such as terrestrial and coastal eutrophication, nitrous oxide emissions, freshwater pollution, and biodiversity loss. Improvement in NUE is associated with several agronomic practices, such as improved irrigation methods, improved fertilizer application considering the 4Rs for nutrient delivery (right product, right rate, right time, and right place), and using hybrid plants with greater productivity and lower nitrogen need. Additionally, nitrogen inhibitors, the split application of nitrogen, irrigation time, and the correct placement method of fertilizer which takes into consideration soil and crop type could improve NUE [93,94,95,96]. Therefore, melatonin must be studied more widely in the agronomic field in order to be considered a possible input into the NUE strategy [12,16,97], including the effects of enhancement by melatonin in the assimilation and metabolization of nitrogen, especially in symbiotic plants [54,55]. It is also important to study the effect of melatonin on rhizosphere and its microbiome, on which there are already some promising data [87,88,89,98,99].

## Figures and Tables

**Figure 1 ijms-23-15217-f001:**
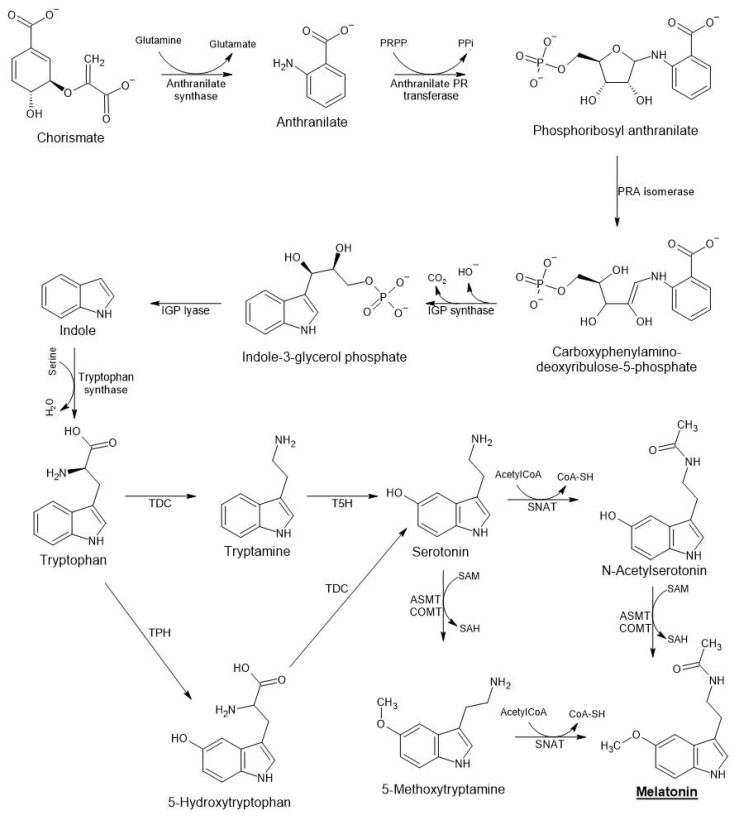
Pathways of melatonin biosynthesis in plants.

**Figure 2 ijms-23-15217-f002:**
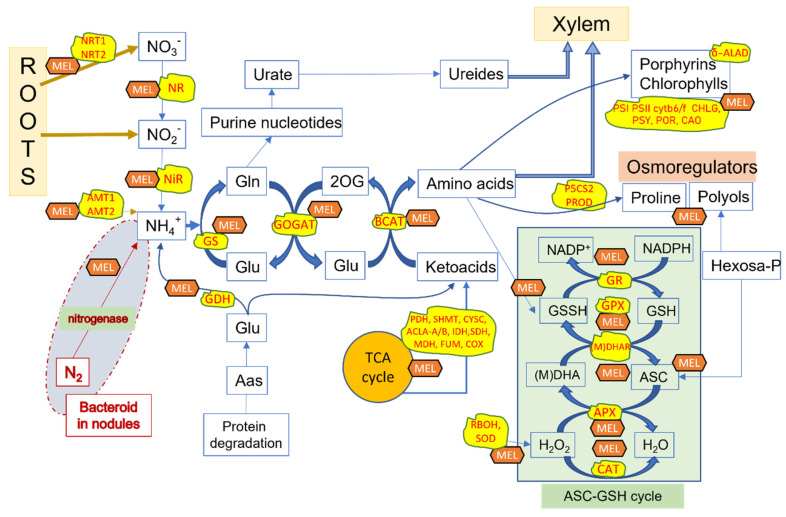
Regulation of nitrogen metabolism and related-pathways by melatonin. Melatonin up-regulates several enzymes and other factors related to the biosynthesis of amino acids in TCA cycles and nitrogen-related reactions, also in an ASC-GSH cycle and osmoregulatory-compound biosynthesis. The different up-regulated genes by melatonin are in the yellow boxes (see Abbreviation Section).

**Table 1 ijms-23-15217-t001:** Effects on nitrogen metabolism by melatonin in different species.

Plant Species	Nitrogen Nutrition/Stress	Melatonin Treatment (µM)	Observed Effects	Reference
Cucumber	Normal High temperature	100	↑ temperature tolerance, NR, GS, GOGAT, GDH, nitrate, ammonium restrained	[47]
	Nitrate: N-excess	100	↑ tolerance, growth, NPK balance, Ca, NR, GS, GOGAT, GDH ↓ damage, nitrate, ammonium	[45]
	Nitrate: N-excess	2	↑ tolerance N excess, co-action with NO, lateral roots, root length, Ca, Mg, Fe, melatonin, NO, IAA, ABA, transcription levels of several genes of N metabolism, IAA, ABA and melatonin	[46]
Apple	Normal (urea) Drought stress	100	↑ drought tolerance, growth, photosynthesis, stomatal open, chls, RWC, NR, NiR, GS, GOGAT, N uptake genes (AMTs, NRTs), N, P, K, Ca, Mg, Cu, Zn, and B levels	[50]
Alfalfa	Nitrate: N-excess	100	↑ tolerance N excess, shoot height, leaves (length, width, area), P, ATP, biomass, amino acids, energy charge, upregulates NR, GS, GOGAT, GDH ↓ total N, nitrate, ammonium, Na, K, Ca	[48]
Wheat	Nitrate and ammonium: N-low	1	↑ N and nitrate, N absorption, N metabolism, NR, GS, growth, yield, in shoots and roots	[51]
Maize	Normal	100	↑ nitrite, nitrate, NR, NiR, GS, GOGAT, GDH ↓ ammonium	[52]
Soybean	Normal Salt/drought stress	50–100	↑ stress tolerance, growth, seed yield and fatty acid; up-regulates cell division, photosynthesis, carbohydrate, fatty acid, and ASC genes	[53]
	Normal	100	↑ number and size of nodules, fresh shoot biomass in 3 varieties	[54]
	Nitrate and ammonium: N-excess, N-normal and N-low	100	↑ tolerance N-excess, N content in N-low, stem diameter, leaf area, nodule number, ATP, biomass in three-N conditions, antioxidant enzymes at N-excess, N-related genes	[49]
	Nitrate and ammonium: N-low	100	↑ nodule number, total N fixed, tolerance to N deficiency, upregulating genes: NR2, NiR, GS1β, GOGAT, AAP6a, promoting enzyme activity: NR, GS, GOGAT, GDH, amino acids, protein, total N, chls, seed yield	[55]
	Normal Drought stress	100	↑ N, NR, NiR, NRT, GS, GOGAT, GDH, protein, proline, ureides, N transport, growth, biomass	[56]
	Normal Drought stress	100	↑ stress tolerance, growth, seed yield, amino acids, photosynthesis, antioxidants, regulates C/N ratio, and plant hormone levels	[57]

↑ Increased content or action; ↓ Decreased content or action.

**Table 2 ijms-23-15217-t002:** Effect of melatonin on enhanced yield in different species.

Plant Species	Melatonin Treatment (µM)	Reference
Rice	50–200	[70]
Wheat	10–500	[71]
	1000	[72]
	50–500	[73]
Moringa	100	[74]
Pomegranate	100	[75]
Maize	10–1000	[52]
Broccoli	20–80	[76]
Kiwifruit	50–200	[77]
Sweet cherry	50–500	[78]
Banana	40–80	[79]
Pepper	100	[64]
Pear tree	100	[80]
Radish	50–300	[81]
Rapeseed	500	[82]
Tomato	100	[83]
Soybean	50–100	[53,55]

## Data Availability

Not applicable.

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
