# Peer review of "Role of Melatonin and Nitrogen Metabolism in Plants: Implications under Nitrogen-Excess or Nitrogen-Low"

_ijms, 2022, doi:10.3390/ijms232315217_

Round 1
Reviewer 1 Report
The paper entitled 'Melatonin and nitrogen metabolism: implications under N-excess or N-low in plants' by Arnao et al. deals with an important topic which is melatonin.
The paper is well written nevertheless I have attached a file containing some comments and suggestions for the author in order to add further value to the work
here below are some general comments
Abstract: in my opinion the abstract lacks 1-2 sentences regarding where future works should focus based on the information gathered in this manuscript et al.
Introduction: the problematic was not properly presented in the introduction section
the lack of information existing in previous work on melatonin and the place of this work among others should be clearly stated in 1-2 sentences or a short paragraph
3 Role of melatonin in nitrogen metabolism: be consistent regarding the titles and subtitles
Table 1: is there any relation between the used concentration and the observed effect?
4 Melatonin in osmoregulation and redox network: be consistent regarding the titles
5 Melatonin, nitrogen, and implications in crops:
6. Conclusions & perspectives: In my opinion, this ssction needs 1-2 section regarding the need for future studies where future works on the same topic should focus.
7 Reference: This sction is fine and many references are very recent
Author Response
Rev 1
The paper entitled 'Melatonin and nitrogen metabolism: implications under N-excess or N-low in plants' by Arnao et al. deals with an important topic which is melatonin.
The paper is well written nevertheless I have attached a file containing some comments and suggestions for the author in order to add further value to the work here below are some general comments
1 Abstract: in my opinion the abstract lacks 1-2 sentences regarding where future works should focus based on the information gathered in this manuscript et al.
R: In this regard, we have added a sentence concerning the future approach.
2 Introduction: the problematic was not properly presented in the introduction section the lack of information existing in previous work on melatonin and the place of this work among others should be clearly stated in 1-2 sentences or a short paragraph
R: We have added a paragraph explaining the problems analyzed in this review.
3 Role of melatonin in nitrogen metabolism: be consistent regarding the titles and subtitles
R: Thank you for your coherent suggestion. We accept the suggested title finally being as:
Role of melatonin and nitrogen metabolism in plants: implications under nitrogen-excess or nitrogen-low
Table 1: is there any relation between the used concentration and the observed effect?
R: Thank you for your annotation. In principle we cannot establish dose: response relationship, although if we look at the Table 1, practically all the studies were carried out with 100 μM melatonin, a very generalized dose for many of the treatments with melatonin in many physiological studies.
4 Melatonin in osmoregulation and redox network: be consistent regarding the titles
R: You will have to blame us, but we do not quite understand the suggestion made by Rev.1 about this section. In it, we present the role of melatonin as a stimulator of the osmoprotective response to stress and present a generally accepted scheme in which melatonin has a central role in the redox network, modulating the response to scavenge and neutralize several ROS and RNS, as well as the expression of transcripts of several antioxidative enzymes (see Figure 2).
5 Melatonin, nitrogen, and implications in crops:
R: Something similar to the first point happens to us with this one. We don't know what the suggestions would be. In the second paragraph, Rev.1 mentions an attachment that we have not located; maybe it is that file where the suggestions for improvement appear.
- Conclusions & perspectives: In my opinion, this section needs 1-2 section regarding the need for future studies where future works on the same topic should focus.
R: Thank you for your suggestion. We have added a more explicit paragraph making suggestions on the necessary studies that should be carried out to address the NUE objectives.
7 Reference: This section is fine and many references are very recent
Thank you for the comment.
Reviewer 2 Report
This work has reviewed the roles and effects of melatonin on plant nutrition and metabolism. However, the language of this manuscript needs to be very carefully and intensively revised before considering being published.
1. Using ‘N’ to roughly represent both nitrogenous compounds (e.g., N compounds) and nitrogen element throughout the manuscript is not recommended. Some of their use may be avoided.
2. Line 8, appearance of ROS and RNS is inevitable if they are aerobic organisms. It is recommended to rephrase to ‘excessive appearance’
3. Lines 9-10, please correct the grammar.
4. Lines 14-15, please consider adjusting the poor combination of subordinate clauses.
5. Lines 30-31, lines 58-59, line 62, lines 111-112, lines 117-118, lines 176-177, lines 205-206, lines 214-215, lines 217-221, please considering rephrase these sentences.
6.Line 92, is the total N levels fully covered by nitrate and ammonium?
7.Line 93, is P necessary in the scope of energy?
8.Line 122, can you define ‘enzyme transcripts’? please also cite the source where this description/phrase is used in the text.
9.Lines 171-172, a sudden appeared italic font?
10.Please unify the format of Journal name and the addition of DOI in the references list.
Author Response
REV 2
This work has reviewed the roles and effects of melatonin on plant nutrition and metabolism. However, the language of this manuscript needs to be very carefully and intensively revised before considering being published.
R: Thank you for your suggestions. The text has been extensively revised and many phrases changed.
1. Using ‘N’ to roughly represent both nitrogenous compounds (e.g., N compounds) and nitrogen element throughout the manuscript is not recommended. Some of their use may be avoided.
R: Thank you for your annotation. We agree that the term "N" is confusing, so we have chosen to delete it and that the full name appears in the text.
2. Line 8, appearance of ROS and RNS is inevitable if they are aerobic organisms. It is recommended to rephrase to ‘excessive appearance’
R: We incorporate the suggestion.
3. Lines 9-10, please correct the grammar.
R: Sentence corrected.
4. Lines 14-15, please consider adjusting the poor combination of subordinate clauses.
R: Sentence corrected.
5. Lines 30-31, lines 58-59, line 62, lines 111-112, lines 117-118, lines 176-177, lines 205-206, lines 214-215, lines 217-221, please considering rephrase these sentences.
R: All the sentences have been rewritten.
6.Line 92, is the total N levels fully covered by nitrate and ammonium?
R: In this study, nitrate and ammonium levels, not other N compounds, were estimated.
7.Line 93, is P necessary in the scope of energy?
R: The authors value the contents of P and ATP as energy markers.
8.Line 122, can you define ‘enzyme transcripts’? please also cite the source where this description/phrase is used in the text.
R: The term is not fortunate. We have rewritten the sentences where it appears.
9.Lines 171-172, a sudden appeared italic font?
R: We have amended the error.
10.Please unify the format of Journal name and the addition of DOI in the references list.
R: We have used the Zotero reference manager and the IJMS-MDPI file that is downloaded online. However, we prefer that the incorporation of the text to the MDPI format is done by a professional, we have problems with it.
Reviewer 3 Report
Thank you for submitting your manuscript to the International Journal ofMolecular Sciences. Generally, the topic fits into the scope of the journal,
however, there are several issues that require fundamental revision. First of
all, the manuscript must be submitted in the layout template formate of MDPI,
and it must respect Scientific Best Practice in terms of the structure
(Introduction/Literature Review - Materials and Methods - Results - Discussion -
conclusions). The materials and methods section is missing and it is mandatory
to be added. In the literature review, it is important that the scientific novelty of the work is established through a critical analysis of related literature. With this, followng questions must be clarified: How does the present work contribute towards the gaps identified? How does it improve upon previous work? Thus, the main questions of the reviewer
are: What is the scientific motivation for the study? What is your scientific hypothesis that you wish to answer with the investigation? Putting the scientific motivation will also help you to identify the novelties that characterises a scientific publication. Moreover, the scope
of the manuscript is unclear. The methdology section is missing and must be added. This is also true for
review papers. I can only assume that the present manuscript is a review paper,
however, there is not any information about this. For the methodology section,
I strongly recommend to include a flow chart illustrating the steps of the
methodology in the beginning of the methodology section. After this, all
applied scientific methods need to be explained in detail.
The results section is missing and must be added. For all figures must be
added the source. The discussion section is missing as well and must be added. In the conclusions, in addition to summarising the actions taken and results, please strengthen the explanation of their significance. It is recommended to use quantitative reasoning comparing with appropriate benchmarks, especially those stemming from previous work.
Author Response
REV 3
Thank you for submitting your manuscript to the International Journal of Molecular Sciences. Generally, the topic fits into the scope of the journal, however, there are several issues that require fundamental revision. First of all, the manuscript must be submitted in the layout template formate of MDPI, and it must respect Scientific Best Practice in terms of the structure (Introduction/Literature Review - Materials and Methods - Results - Discussion - conclusions). The materials and methods section is missing and it is mandatory
to be added. In the literature review, it is important that the scientific novelty of the work is established through a critical analysis of related literature. With this, followng questions must be clarified: How does the present work contribute towards the gaps identified? How does it improve upon previous work? Thus, the main questions of the reviewer are: What is the scientific motivation for the study? What is your scientific hypothesis that you wish to answer with the investigation? Putting the scientific motivation will also help you to identify the novelties that characterises a scientific publication. Moreover, the scope of the manuscript is unclear. The methdology section is missing and must be added. This is also true for review papers. I can only assume that the present manuscript is a review paper, however, there is not any information about this. For the methodology section, I strongly recommend to include a flow chart illustrating the steps of the
methodology in the beginning of the methodology section. After this, all applied scientific methods need to be explained in detail.
The results section is missing and must be added. For all figures must be added the source. The discussion section is missing as well and must be added. In the conclusions, in addition to summarising the actions taken and results, please strengthen the explanation of their significance. It is recommended to use quantitative reasoning comparing with appropriate benchmarks, especially those stemming from previous work.
R: We want to understand that there has been an error in the interpretation of our work, since it is a review, and Rev.3 makes suggestions as if it were an experimental article, suggesting sections such as Material and Methods, Results,...
Reviewer 4 Report
1. need more references to support discussion
2. please check the references according to the templete
3. please make figure 3 clear about color or size of text, need reduse it and change color green box to others color to make more clear
4. figure 3 or figure 2, please check again
5. Table alredy according to templete please check again
6. 7. plagiarism 24%

Author Response
REV 4
- need more references to support discussion
R: We have incorporated several more references in the new paragraphs.
- please check the references according to the templete
R: We have used the Zotero reference manager and the IJMS-MDPI file that is downloaded online. However, we prefer that the incorporation of the text to the MDPI format is done by a professional, we have problems with it.
- please make figure 3 clear about color or size of text, need reduse it and change color green box to others color to make more clear
R: The suggested changes have been made.
- figure 3 or figure 2, please check again
R: Sorry for the mistake. Figure 3 does not exist.
- Table already according to templete please check again
R: We prefer that the incorporation of the text to the MDPI format is done by a professional, we have problems with it.
- plagiarism 24%
R: The coincidences must be given mainly with our papers and with the references.
See pdf attached
R: We have corrected or incorporated all the suggestions that appear in yellow in the pdf file.
Reviewer 5 Report
Dear Authors
I read the article, and it was an interesting subject.
The general comments
article dose does not have an MDPI format!! Please check it
and let me know what is the novelty of this study.
Why did you use self-citation?
Author Response
REV 5
I read the article, and it was an interesting subject.
R: Thank you for your review and feedback.
The general comments article dose does not have an MDPI format!!
R: We prefer that the incorporation of the text to the MDPI format is done by a professional, we have problems with it.
Please check it and let me know, what is the novelty of this study.
R: Many data pointed to the great potential of melatonin to use to improve nitrogen fertilization in plants, mainly to the NUE strategy [12,16,97], including the effects of enhancement by melatonin in the assimilation and metabolization of nitrogen, especially in symbiotic plants [54,55]; also delve into studies on the effect of melatonin on rhizosphere and its microbiome, where there are some promising data [87–89,98,99].
Why did you use self-citation?
R: We are a research group with 25 years of experience in the field of melatonin, our work in this regard exceeds one hundred so we believe it is relevant to cite some of our papers in this review, as they are a benchmark for other colleagues.
Round 2
Reviewer 2 Report
line 170, is the italic font correct?
In the reference list, some articles' DOI is given, while others are not. Please ensure it is aligned with the current journal's guidelines.
Reviewer 3 Report
The revisions have not been done.
Moreover, the authors had a substantial misunderstanding what means to write a scientific review article. The reviewer understood very well that this shall be a review manuscript, however the authors didnt understand how to write a scientific review article. Of course, it is scientific best practice also for a review article to describe the methdology, the results and to discuss them.
I provide again the chance for a substantial revision. As the authors obviously have no idea what means a scientific review article and how to write for a review article a methodology section, I recommend to have a look into samples to become familiar with the methodology for writing a review paper:
https://www.mdpi.com/2071-1050/12/15/6274
https://www.mdpi.com/2073-4441/13/2/225
https://www.mdpi.com/2073-4441/14/12/1834
https://www.mdpi.com/2071-1050/14/9/5503
Author Response
We included a new section:
- Methodology
In this study, a systematic literature review is conducted on the role of melatonin in the nitrogen metabolism of plants. The bibliometric analysis was conducted in three stages: i) defining the keywords; ii) selecting database, and iii) searching relevant articles, and analyzing data. Peer-reviewed publications were searched covering the period 1995, the year melatonin was first identified in plants, and 2022. The search was performed in the title and keywords of the publications, selecting English and other languages, and articles in journals and book chapters, both experimental and review types, which were related to melatonin and nitrogen metabolism in plants. A systematic database search of peer-reviewed articles was conducted using the Science Citation Index Expanded (SCI-Expanded) database of the Web of Science from Thomson Reuters. Also, Scopus, Google Scholar and PubMed database were used. Our Plant Hormones & Development research group from University of Murcia has a great experience in melatonin and for 25 years has been generating a database integrated by some 4,000 references that include all the works related to melatonin in plants. The results indicated that only 75 references were registered, of which 62 are from the last five years.
Reviewer 5 Report
Dear Authors
You did not use the MDPI format!! I can not accept your article.
best
Author Response
The IJMS-MDPI format has been included.
Round 3
Reviewer 3 Report
A minium part of my comments was considered, namely only the scientific duty to add the methdology section.
The Manuscript still has big flaws, particularly in terms of own contributions. In the present form it is a collection of literature, however own discussions and interpretations are missing. As this fundamentally characterises a scientific publication, the manuscript is still far from being publishable. The sources of the figures are continuosly missing.
In the introduction was written: "In addition, the possibilities of using melatonin in crops for more efficient uptake, assimilation and metabolization of nitrogen from soil, and the implications for Nitrogen Use Efficiency (NUE) strategies to improve crop yield were also discussed, i.e., to increase crop yields with suboptimal levels of nitrogen was proposed". However, there is not any discussion beside the cited references, even the discussion section is completely missing.
The conclusions are vague, without a real content.
Examples:
In relation to nitrogen nutrition, melatonin improves growth, survival rate, and stress tolerance (of what, which kind of stress?). In nitrogen-excess situations (what exactly means in this context excess?), melatonin increased nitrogen-stress tolerance, reducing nitrate/ammonium uptake (to which extent?) and up-regulating nitrogen-related genes resulting in a rebalancing of nitrogenous compounds, redirecting amino acid, proline and ureide levels (to which extent?). Under nitrogen-low conditions (what exactly means this in this context?), an improvement in the levels of nitrogen compounds was observed (to which extent?), with a greater absorption of nitrogen and an increase in levels of amino acids, proteins, and chlorophylls (to which extent?). In some cases of abiotic stress, such as drought or high temperatures, melatonin regulated nitrogen-related genes, optimizing osmoregulation response, mineral uptake, and total nitrogen levels in the tissues (to which extent?). In most of the cases studied, melatonin increased growth and plant biomass (to which extent?). In soybean cultivation, melatonin improves yields possibly due to the stimulation of the number and size of root nodules.
Reviewer 5 Report
Dear author
Let me know what is your goal for this article.
what is the novelty of this article?
why the author used self-citation?
it is not clear to me!
best